# Localized Surface Plasmon-Enhanced Infrared-to-Visible Upconversion Devices Induced by Ag Nanoparticles

**DOI:** 10.3390/ma16051973

**Published:** 2023-02-28

**Authors:** Yuyi Zhang, Chengjun Liu, Xingyu Liu, Ziyu Wei, Hui Tao, Feng Xu, Lixi Wang, Jiangyong Pan, Wei Lei, Jing Chen

**Affiliations:** 1Joint International Research Laboratory of Information Display and Visualization, School of Electronic Science and Engineering, Southeast University, Nanjing 210018, China; 2School of Electronic and Information Engineering, Nanjing University of Information Science & Technology, Nanjing 210044, China

**Keywords:** upconversion device, quantum tunneling, localized surface plasmon (LSP)

## Abstract

Upconversion devices (UCDs) have motivated tremendous research interest with their excellent potential and promising application in photovoltaic sensors, semiconductor wafer detection, biomedicine, and light conversion devices, especially near-infrared-(NIR)-to-visible upconversion devices. In this research, a UCD that directly turned NIR light located at 1050 nm into visible light located at 530 nm was fabricated to investigate the underlying working mechanism of UCDs. The simulation and experimental results of this research proved the existence of the quantum tunneling phenomenon in UCDs and found that the quantum tunneling effect can be enhanced by a localized surface plasmon.

## 1. Introduction

Numerous research studies on upconversion devices (UCDs) have been reported, owing to their application in photovoltaic sensors, semiconductor wafer detection, biomedicine, and light conversion devices, especially devices that can turn near-infrared (NIR) light to visible light [1,2,3,4,5]. Early NIR UCDs were integrated with infrared photodetectors and visible light emitters, which involve complicated steps [6,7,8]. This is because the InGaAs detector and the Si-based readout circuit grow on different substrate materials, which need to be prepared separately, and then need to be connected one-to-one through an indium column. However, since such NIR UCDs have some disadvantages, such as lattice mismatch (which can occur between the materials comprising the detector and the materials comprising light emitting diode) and the inability to prepare an NIR UCD in a large area (owing to the high costs of epitaxial grown inorganic devices) [9], other design structures of NIR UCDs have been considered [9,10], The appearance of devices that could directly convert NIR light to visible light have greatly expanded the application of NIR UCDs. These types of devices have the advantages of fast response, simple structure, and convenient preparation [11]. Ban et al. first raised the concept in 2007 of integrating InGaAs/InP photodetectors and organic light-emitting diodes (OLEDs) directly into a new device [12]. Kim et al. have improved this device by changing the materials of each layer [2]. Unfortunately, there were too many layers in this new device structure, which made it impossible to complete carrier transfer efficiently. Many reports have indicated that metal nanoparticles could be helpful in carrier transport; thus, adding metal nanoparticles such as silver (Ag) is of great significance for the improvement of UCDs [13,14]. Consequently, Zhou et al. prepared similar devices in 2020, which marked great progress in UCDs [10].

The addition of Ag nanoparticles (NPs) is particularly important for the upconversion of NIR light while studying these devices. Metal nanoparticles have a great influence in the quantum domain due to their size, such as the quantum size effect and quantum tunneling effect. The unique properties of metal nanoparticles ensure they have promising applications in optical, electrical, biological, and chemical catalysis [15,16,17]. Previous experiments have also found that Ag NPs feature good optical and electrical properties [18], such as enhancement of Raman spectra by Ag nanoparticles [19] and increased device conductivity owing to the good carrier transport characteristics of Ag [20]. This paper will focus on a more noteworthy property, the localized surface plasmon effect produced by Ag NPs, which greatly improves the performance of UCDs [13,14]. Moreover, relevant studies have shown that gold (Au) nanoparticles can generate hot hole-electron pairs under illumination [21]. Therefore, the existence of Ag NPs should also produce hole-electron pairs, which to some extent increases the current density and electric field strength, making quantum tunneling more obvious [22]. Because of these excellent properties, various devices containing Ag NPs have developed rapidly. In 2020, an array of Ag NPs were designed which created Localized Surface Plasmon Resonance (LSPR) near the surface of silicon (Si) [14]. In another study, Yang et al. found that the lifetime was extended when the surface plasmon resonant peak of silver nanoparticles was adjusted to resonate with the Q- absorption band [23].

In the current research, a device structure that can turn NIR light located at 1050 nm into visible light located at 530 nm was fabricated to explain the tunneling mechanism in UCDs, which is similar to a switch sensitive to infrared light. The localized surface plasmon formed by the effective deposition of Ag NPs has proved to be possible through theoretical calculation, such that it can enhance the quantum tunneling effect in UCDs. The device characterization results verified the tunneling effect caused by Ag NPs under NIR light. In previously reported devices, excitons were recombined by holes generated in the photosensitive layer. After adding Ag NPs, holes could be injected from the ITO electrode, which greatly improved the performance of the device. The current research is significant in improving the performance of similar devices by understanding the working mechanism of UCDs. It may have a far-reaching impact on the preparation of optoelectronic devices.

## 2. Materials and Methods

Materials: Magnesium acetate tetrahydrate (Sigma-Aldrich, St. Louis, MO, USA, 99.99%), Zinc acetate dihydrate (Sigma-Aldrich, 99.99%), Tetramethyl ammonium Hydroxide (Sigma-Aldrich, >97%), Ethanol (H_2_O ≤ 50 ppm (by K.F.), Meryer, Shanghai, China, 99.5%), Methyl sulfoxide (H_2_O ≤ 50 ppm (by K.F.), Meryer, 99.7%), Silver nitrate (Sigma, 99.99%), Oleylamine (Aladdin, 80–90%), Toluene (Sinopharm, Beijing, China, >99.5%), Lead monoxide (Aladdin, 99.999%), Oleic acid (Adamas, >90%), 1-Octadecene (Macklin, >95%), Acetone (Sinopharm, >95%), Hexamethyldisilathiane (Energy chemical, 98%), Tetrabutyl ammonium iodide (Maya, >99%), Methanol (Acros, 99.9%), 1,2-Ethanedithiol (Sigma-Aldrich, >99%), Acetonitrile (Sigma-Aldrich, 99.8%), Molybdenyl acetylacetonate (Adamas, 99%), Poly [4,4′-(N-(4-secbutylphenyl)diphenylamine] (p-OLED, MW > 40,000), CdSe/ZnS quantum dots (Mesolight), Nafion (Sigma-Aldrich), Isopropanol (Sinopharm, >99.7%)

About manufacturer and country from where the equipment was sourced, they are shown in Appendix A in supporting information.

Synthesis of Zn_0.95_Mg_0.05_O: We dissolved 0.1 mol of tetramethyl ammonium hydroxide (TMAH) in ultra-dry ethanol, 0.02 mol of zinc acetate and 0.00105 mol of magnesium acetate in dimethyl sulfoxide (DMSO), and mixed the two solutions vigorously for 1 h at room temperature to synthesize Zn_0.95_Mg_0.05_O nanoparticles. The suspension was washed 3 times with absolute ethanol/acetone (1:3 volume ratio). Finally, the nanoparticles were dispersed in ultra-dry ethanol, about 30 mg/mL. (The suspension was centrifuged at 4000 r.p.m. for 5 min.)

Synthesis of PbS Quantum Dots(QDs): We dissolved 0.9 g (4.04 mmol) of lead oxide in 9 mL of mixed solvent consisting of oleic acid (OA)(3 mL) and 1-Octadecene (ODE) (6 mL). Then the mixed solution was heated under N_2_ flow at 95 °C for 4 h, after which an additional 10 mL of ODE was added for further dilution. After that, 420 µL of hexamethyldisilathiane (TMS) was dissolved in another 10 mL of ODE. The 10.42 mL of TMS solution was then injected into the reaction solution at 122 °C, accompanied by vigorous stirring. We removed the heating jacket immediately after 30 s and used an ice water bath to cool down the mixed solution. Finally, while keeping the temperature of the solution at 20 °C, we isolated the nanocrystals by adding acetone. The obtained solid was dispersed in 12 mL of toluene for purification, and was precipitated again with acetone/hexane (1:4 volume ratio). The step above was repeated three times. Then it was precipitated once with acetone/octane (1:5 volume ratio), and finally dispersed in octane. (The suspension was centrifuged at 8500 r.p.m. for 5 min.) X-ray diffraction (XRD) spectra for structural analysis are shown in Appendix A.

Synthesis of Ag NPs: We dissolved 0.1359 g of AgNO_3_ (0.8 mmol) in 3.2 mL of OLA and then add the mixed solution above into 80 mL of toluene. The solution was heated to 110 °C under a N_2_ flow for 6 h, accompanied by vigorous stirring. After the reaction, the heating jacket was removed and the solution was cooled to room temperature by ice water bath. Finally, the Ag NPs were isolated by adding anhydrous ethanol. The obtained solid was washed with toluene/ethanol (1:4 volume ratio) twice, and then dissolved in toluene [10]. (The suspension was centrifuged at 6000 r.p.m. for 5 min.) XRD spectra for structural analysis are shown in Appendix A.

Synthesis of MoOx Precursor: In the glove box filled with nitrogen, 20 mg of Molybdenyl acetylacetonate (MoO_2_(acac)_2_) was dissolved in a mixed solvent (1 mL) consisting of isopropanol and 1-butanol (1:1 volume ratio). The mixed solution was stirred on a hot plate at 60 °C for 1 h. Nafion was also diluted with a mixture of isopropanol and 1-butanol (1:1 volume ratio) to obtain a concentration of 5 wt%. Before spin coating, we stirred the two mixed solutions above for 20 min and dilute them 50 times. The final standard concentration of MoO_2_(acac)_2_ was 0.4 mg/mL, and the standard concentration of Nafion was kept at 0.1 wt% [24].

Simulated Software: Computer simulation technology (CST)

Device Fabrication: The structure of UCD was ITO/Zn_0.95_Mg_0.05_O/Ag NPs/PbS QDs/MoOx/poly-TPD/(CdSe/ZnS) QDs/Zn_0.95_Mg_0.05_O/Ag electrode. ITO glass was cleaned using deionized water, acetone and isopropyl alcohol in that order in an ultrasonic bath for 10 min each, followed by ultraviolet treatment for 30 min. The substrates were then transferred to an N_2_-filled glove box. In this environment, Zn_0.95_Mg_0.05_O nanoparticles were spin-coated on top of the pretreated ITO glass at 3000 r.p.m. for 30 s, followed by heat treatment at 70 °C for 10 min. Ag NPs (5 mg/mL) dissolved in octane were spin-coated on the Zn_0.95_Mg_0.05_O layer at 3000 r.p.m. for 30 s, followed by heat treatment at 70 °C for 5 min. Then, the PbS layer was deposited via layer-by-layer spin coating. For each layer, 400 µL of PbS solution were spin-cast onto the Zn_0.95_Mg_0.05_O substrate at 2000 r.p.m. for 15 s. A tetrabutylammonium iodide solution was then dropped onto the film and left for 30 s, followed by two rinse–spin steps using methanol. After that, a MoO_x_ precursor was spin-coated at 3000 r.p.m. and annealed at 70 °C for 15 min, and an electron-blocking layer was spin-coated onto the MoO_x_ substrate at 4000 r.p.m. for 60 s using poly-TPD solution, followed by heat treatment at 80 °C for 10 min. Then CdSe/ZnS QDs dissolved in octane was spin-coated at 2000 r.p.m. for 30 s, followed by heat treatment at 80 °C for 10 min. Afterwards the Zn_0.95_Mg_0.05_O film was spin coated at 2000 r.p.m. for 30 s, followed by heat treatment at 80 °C for 10 min. Finally, 120 nm Ag was deposited.

Characterizations: The absorption spectra were measured using a UV-Visible instrument (U-4100). The photoluminescence (PL) spectra were measured using an NIR spectrometer (Maya2000). The current-voltage-luminance (I-V-L) and current density-external quantum efficiency-luminance (J-EQE-L) characteristics were characterized using a Photo Research (PR670) spectrometer equipped with a source meter (Keithley 2400). The chemical compositions and film morphology were analyzed with an energy-dispersive spectrometer (EDS)-equipped field-emission scanning electron microscope (JEOL-7800F).

## 3. Results and Discussion

Firstly, the prepared quantum dot material was characterized to confirm whether it could meet the requirements of the further device preparation. Figure 1a shows the absorption spectra of PbS QDs located at 1050 nm and Figure 1b shows that the photoluminescence peak of the prepared PbS QDs was 1130 nm with NIR emission. Figure 1c shows the absorption spectra of CdSe/ZnS QDs located at 515 nm and Figure 1d shows that the photoluminescence peak of the prepared CdSe/ZnS QDs was 530 nm with green emission. The UCD mentioned above absorbed NIR light (~1050 nm) and emit green light (~530 nm). Then, the UCD was further prepared based on the two types of QD, i.e., PbS and CdSe/ZnS, and demonstrated that it could turn NIR light into visible light. (Excitation for Figure 1b was infrared light located at 1000 nm whose Peak Width at Half Height was 100 nm. Excitation for Figure 1d was visible light located at 525 nm.)

The SEM and EDS results of Zn_0.95_Mg_0.05_O film modified by the Ag NP layer showed that the elements of Zn, O, and Ag were distributed uniformly (Figure 2), revealing the uniform composition of the film. Furthermore, it was found that there were some small islands of Ag NPs deposited on the surface of Zn_0.95_Mg_0.05_O, as shown in Figure 2.

Our second aim was to understand the mechanism of this UCD. In Figure 3a, the layers distributed between the ITO electrode and evaporation electrode Ag can be seen. From the bottom to the top, the device structure is illustrated as ITO (190 nm)/Zn_0.95_Mg_0.05_O (100 nm)/Ag NPs (20 nm)/PbS QDs (400 nm)/MoO_x_ (15 nm)/poly-TPD (15 nm)/(CdSe/ZnS) QDs (60 nm)/Zn_0.95_Mg_0.05_O (80 nm)/Ag electrode (120 nm). The light-emitting area of this UCD was 4 mm^2^.

To understand the working mechanism of the UCD, it was necessary to analyze the energy level relationship and carrier transfer process among the layers. As shown in Figure 3b, the highest occupied molecular orbital (HOMO) energy levels of Zn_0.95_Mg_0.05_O, PbS, poly-TPD, and CdSe/ZnS were 7.4 eV, 5.6 eV, 5.5 eV, and 6.8 eV, respectively. The lowest unoccupied molecular orbital (LUMO) energy levels of Zn_0.95_Mg_0.05_O, PbS, poly-TPD, and CdSe/ZnS were 3.7 eV, 4.2 eV, 2.3 eV, and 4.3 eV, respectively. The matched energy level of each layer ensured the efficient transfer of carriers in the device [25]. For one part, when PbS film was irradiated by NIR light (1050 nm), hole-electron pairs were generated and then moved under the horizontal electric field. Electrons were passed through the barrier between PbS and Zn_0.95_Mg_0.05_O, leaving holes to transfer from poly-TPD to CdSe/ZnS QDs. Meanwhile, some holes were injected from the ITO electrode and accumulated at the interface between ITO and Zn_0.95_Mg_0.05_O layer. For the other part, electrons were injected from the Ag electrode to CdSe/ZnS QDs through the Zn_0.95_Mg_0.05_O layer, thus electrons and holes could be combined within CdSe/ZnS QDs as excitons to emit light. Therefore, this type of UCD worked as a function to transfer NIR light (1050 nm) to green light (530 nm) in this work.

However, it was noticed that the HOMO level of Zn_0.95_Mg_0.05_O (~7.4 eV) was very far from that of ITO (~4.8 eV). The high valence band (VB) of Zn_0.95_Mg_0.05_O prevents holes from migrating at the interface of Zn_0.95_Mg_0.05_O and ITO, making it difficult for the device to light up [25]. The UCD could only work when the electrons generated from PbS QDs passed through the barrier (~0.5 eV) between PbS and ITO under NIR light; meanwhile the holes generated from PbS QDs were transferred and recombined with electrons injected from the Ag electrode within CdSe/ZnS QDs. This was the key point of this type of UCD: to elucidate the function and switch the device on/off with or without NIR radiation.

Thus, in order to achieve the switching property, the function of Zn_0.95_Mg_0.05_O/Ag NPs could be further studied and the layers of this key part could be analyzed separately. If it was without NIR light, as in Figure 4a, the energy barrier between ITO and Zn_0.95_Mg_0.05_O would be high (~2.6 eV) with a thicker band bend; thus, holes could not easily cross through the barrier. When the NIR light was applied (2 mw/cm^2^), the band bend could be slightly increased [10], and the NIR light would result in pairs of electron holes in the PbS layer. If a positive bias was applied (2~4 V), electrons could be moved and collected at the boundary between the Zn_0.95_Mg_0.05_O and PbS layers, and then pass through the barrier (~0.5 eV), as shown in Figure 4b [9]. In this case, the leaving holes generated within the PbS QDs could be transferred from poly-TPD to CdSe/ZnS QDs, combining with the electrons injected from the Ag electrode [26]. Thus, the upconversion process that switches the NIR light to green emission could be elucidated. However, a strong NIR source power would be necessary to generate more hole-electron pairs of carriers within PbS QDs [9]. In other words, NIR light with low intensity could not make the device work perfectly.

The VB of Zn_0.95_Mg_0.05_O [27] could be significantly bent with the addition of the Ag NP layer, increasing the chance for the holes accumulating and tunneling through the energy barrier at the interface between Zn_0.95_Mg_0.05_O and ITO [28]. In this way, more holes could participate in the formation of excitons, as a result of improving the brightness of the device and reducing the threshold voltage.

Next, in order to investigate the function of the Ag NPs layer in the upconversion process and its feasibility, a model was established to simulate the role of Ag NPs in this process by computer simulation technology (CST). This model was established on the basis of observing the morphology of Ag nanoparticles using a transmission electron microscope (TEM) (shown in Appendix A) which monitored the change of electric field on the film surface after applying near-infrared light. Ag NP islands were randomly generated at different positions in this model to ensure the credibility of the simulation model relative to the real nanoparticles. Through the established model, it was easily found that a force in the form of a localized surface plasmon (LSP) strengthened the electric field between Zn_0.95_Mg_0.05_O film when there was 1050 nm NIR light shining from below, and thus the tunneling was strengthened. The thickness of the Zn_0.95_Mg_0.05_O model was set to 100 nm and the radius of the square-like islands (Ag NPs) above the Zn_0.95_Mg_0.05_O film was 25 nm (see Figure 5a). From the simulation results, the obvious enhancement of the electric field near the Ag NPs islands can be seen (Figure 5b,c), which was consistent with the results of other different devices [29].

After the aforementioned process, the devices mentioned above were prepared in order to further demonstrate the role of Ag NPs. The upconversion devices were simply characterized and it was found that the performance of the device differed greatly with or without infrared light. Under NIR light, the brightness and external quantum efficiency (EQE) of the device were significantly improved, increasing about five times, as shown in Figure 6a,b. On the one hand, the reason for this could be that the photogenerated holes excited by infrared light within the PbS QDs layer were recombined with electrons injected from the cathode. On the other hand, as mentioned above, the enhanced quantum tunneling effect arising from the localized surface plasmon with the addition of Ag NPs were helpful in improving electroluminescence performance. In past research, the current of UCD without Ag NPs could only reach 10 A/m^2^ [2], but in this research, it reached 2500 A/m^2^ at a voltage of 4 V with a testing area of 4 mm^2^, as shown in Figure 6c. To some extent, it was caused by the different materials of the NIR-sensitizing layer, which was SnPc:C_60_ in the paper [2] above and PbS in the current research. However, after the addition of Ag NPs, NIR light in fact formed the localized surface plasmon effect near the Ag NPs island, enhancing the electric field and making tunneling more obvious, allowing more electrons to participate in carrier transport. From Figure 6d, it can be seen that the threshold voltage changes from 4.4 V to 3.4 V as the NIR light switch turns from off to on, which is consistent with the simulation results discussed above. This phenomenon was also due to the fact that the electric field between Zn_0.95_Mg_0.05_O film was strengthened by LSP which contributes to the enhancement of quantum tunneling. Therefore, with the applied voltage maintained at 4 V, a switched UCD could work under excitation of NIR light.

Finally, in order to further prove the electron tunneling mechanism of Zn_0.95_Mg_0.05_O thin films, the current–voltage characteristic curve of ITO/Zn_0.95_Mg_0.05_O/AgNP/PbS QDs/Ag structure was constructed, as shown in Figure 7a. In metal-oxide-metal and metal-oxide-semiconductor systems, the direct electron tunneling effect can occur if the thickness of the intermediate oxide is less than 4 nm; if the thickness exceeds 4 nm, F-N tunneling is observed [30,31]. We fitted the curve according to F-N tunneling theory, using a ZnO layer thickness of ~100 nm and an electric field strength of ~40 mV/mm, as shown in Figure 7b, which is consistent with the reported F-N curve formula [32]. As a result, it was confirmed that the electrons could be successfully transferred from PbS QDs to the Zn_0.95_Mg_0.05_O interface due to energy-bending by the tunneling process. Meanwhile, inducing Ag NPs was beneficial for such a tunneling process due to the LSP effect.

## 4. Conclusions

In this paper, a NIR-to-visible upconversion device (UCD) with switching characteristics was fabricated. The device structure was characterized as ITO/Zn_0.95_Mg_0.05_O/Ag NPs/PbS QDs/MoO_x/_poly-TPD/(CdSe/ZnS) QDs/Zn_0.95_Mg_0.05_O/Ag electrode.

In this structure, we found that, firstly, as PbS film was irradiated by NIR light (1050 nm), hole-electron pairs could be generated and then moved under the horizontal electric field. Electrons could be passed through the barrier between PbS QDs and Zn_0.95_Mg_0.05_O, leaving holes to transfer from poly-TPD to CdSe/ZnS QDs. Meanwhile, holes could be injected from the ITO electrode and accumulated at the interface between the ITO and Zn_0.95_Mg_0.05_O layers. Secondly, we found that electrons were injected from the Ag electrode to CdSe/ZnS QDs through the Zn_0.95_Mg_0.05_O layer; thus, electrons and holes could be combined within CdSe/ZnS QDs as excitons to emit light. Therefore, it was demonstrated that UCD could convert 1050 nm NIR light to 530 nm green light quickly and directly.

It is worth noting that the electrons could be successfully transferred from PbS QDs to Zn_0.95_Mg_0.05_O interface due to energy-bending by the tunneling process, which was proved by fitting the F-N quantum tunneling curve of the voltage and current characteristics. Meanwhile, inducing Ag NPs was beneficial for such tunneling processes due to the LSP effect, which can decrease the threshold voltage and increase the brightness of the device under NIR radiation.

This type of UCD has great application potential in the field of optoelectronic devices because it bring more possibilities for facilitating the complicated barrier interfaces.

## Figures and Tables

**Figure 1 materials-16-01973-f001:**
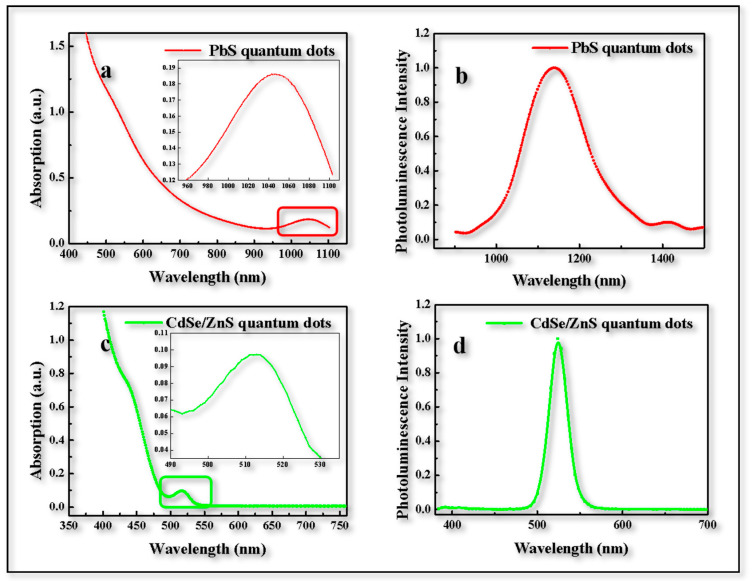
**Quantum dot characterization.** (**a**) Absorption spectra of PbS QD film, inset is detail view of peak position. (**b**) PL spectra of PbS QDs film. (**c**) Absorption spectra of CdSe/ZnS QDs film, inset is a detailed view of the peak position. (**d**) PL spectra of CdSe/ZnS QDs film.

**Figure 2 materials-16-01973-f002:**
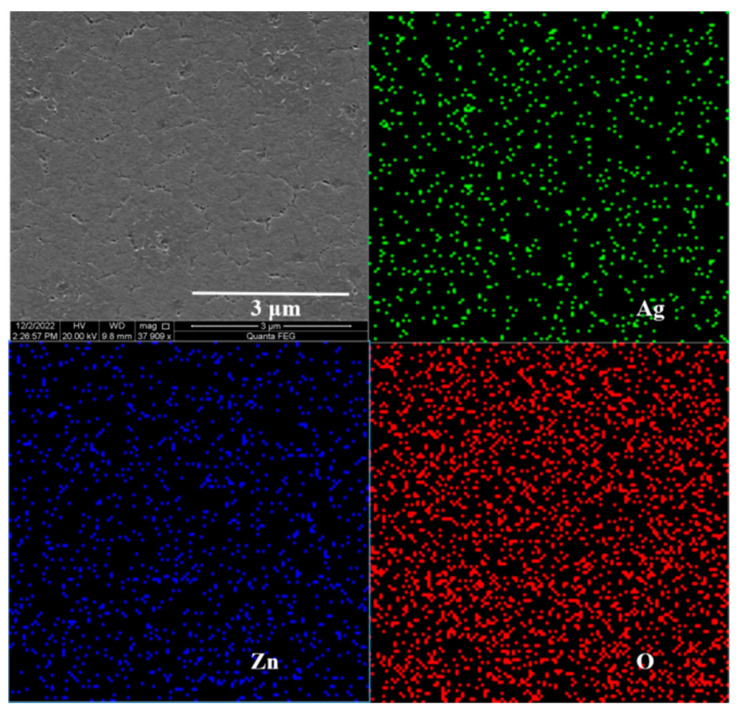
SEM characterization and EDS mapping of Zn_0.95_Mg_0.05_O films modified by Ag NPs.

**Figure 3 materials-16-01973-f003:**
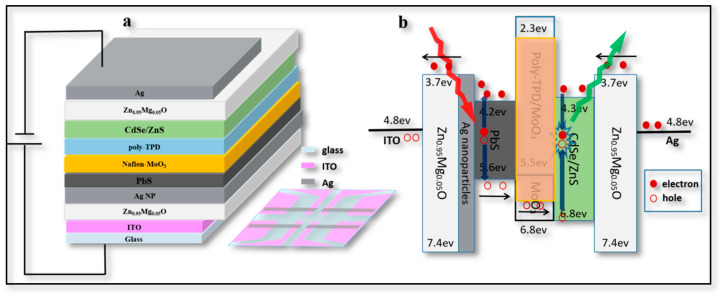
**Structure of UCD.** (**a**) Schematic cross-sectional view of UCD structure, inset is a top view of the device structure. (**b**) Energy level diagram of each film and carrier transfer mechanism.

**Figure 4 materials-16-01973-f004:**
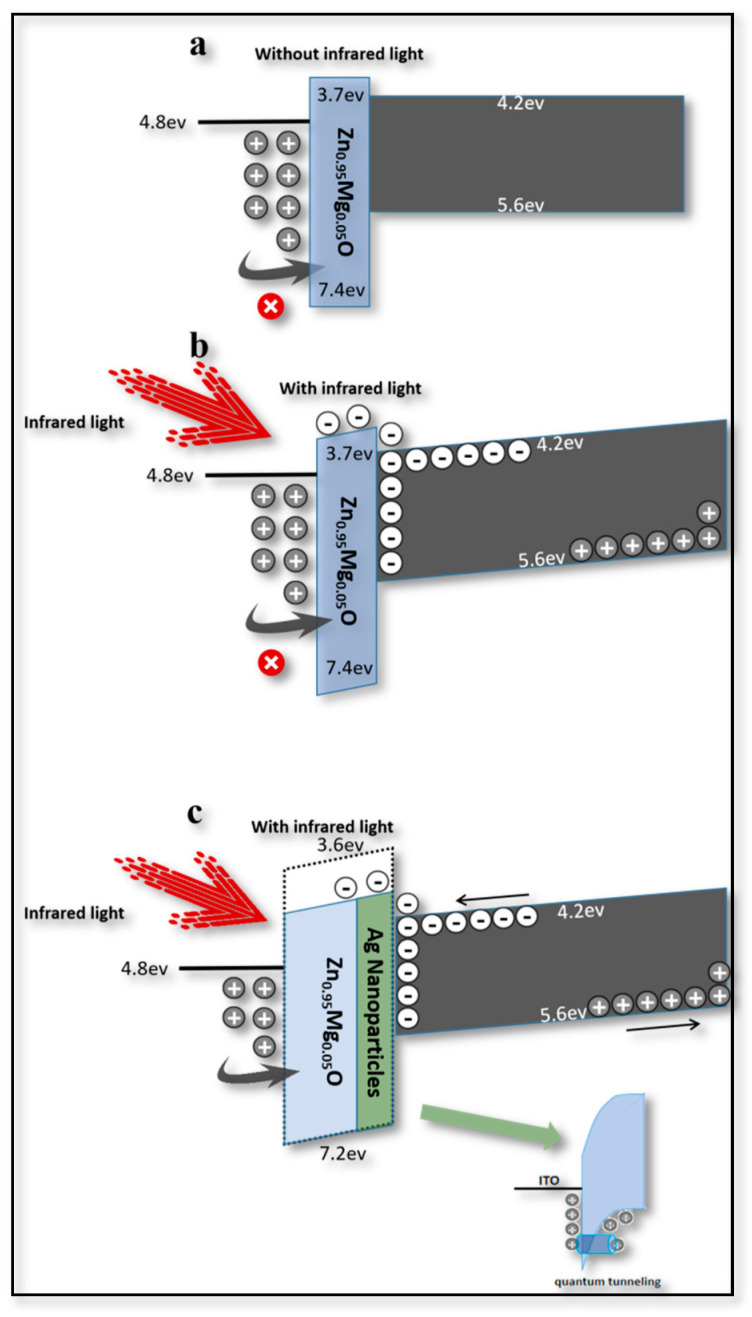
**Potential barrier mechanism diagram of Zn_0.95_Mg_0.05_O and PbS interface.** (**a**) Without Ag NPs and NIR light. (**b**) Without Ag NPs and with NIR light. (**c**) With Ag NPs and NIR light.

**Figure 5 materials-16-01973-f005:**
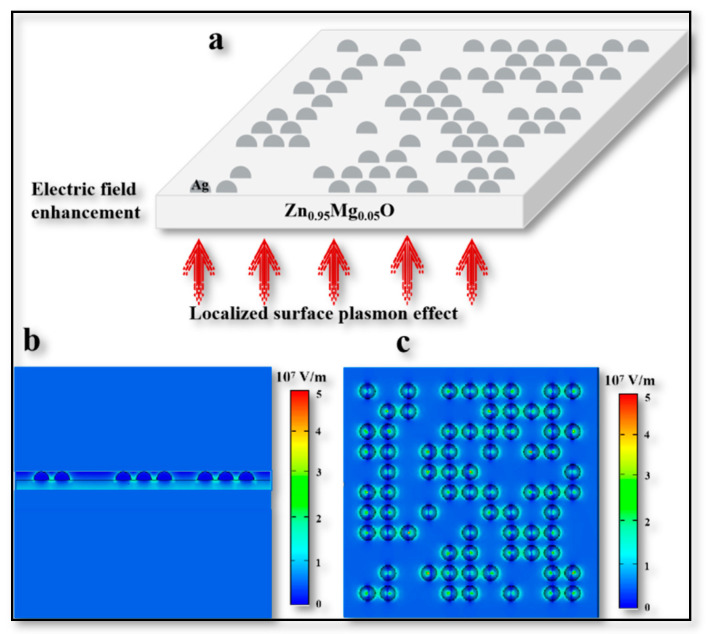
**CST Simulation diagram.** (**a**) Schematic diagram of established simulation model. (**b**) Section diagram of simulation results. (**c**) Vertical view of simulation results.

**Figure 6 materials-16-01973-f006:**
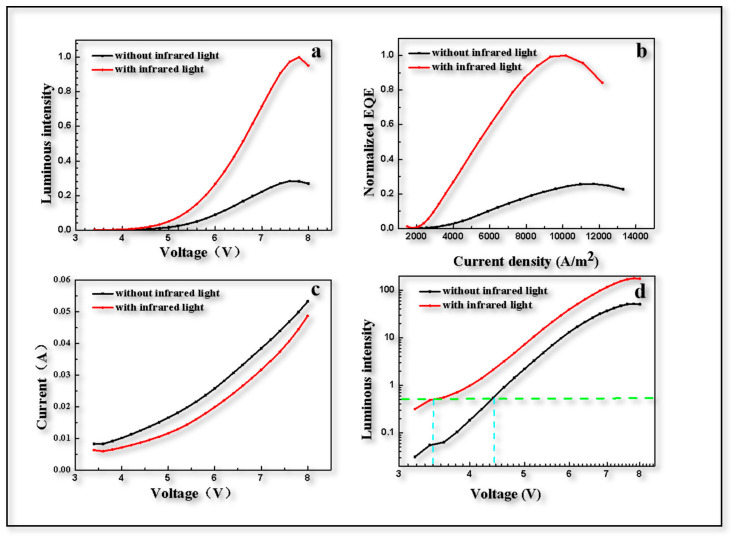
**Photoelectric performance test of upconversion device.** (**a**) Value of green light emission intensity of device with voltage change. (**b**) Normalized EQE varies with current density. (**c**) I-V curve of current with voltage. (**d**) Change of switching on voltage of 530 nm green light.

**Figure 7 materials-16-01973-f007:**
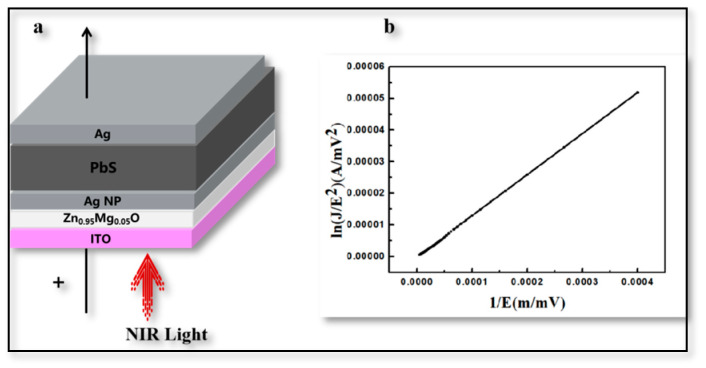
**F-N Fitting Results.** (**a**) Schematic cross-sectional view of tunneling structures. (**b**) Fitting results of ITO/Zn_0.95_Mg_0.05_O/Ag NP/PbS QDs/Ag structure.

## Data Availability

Data is contained within the article or Appendix A. For more information, please contact the author Yuyi Zhang (220211715@seu.edu.cn).

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
