# Peer review of "Localized Surface Plasmon-Enhanced Infrared-to-Visible Upconversion Devices Induced by Ag Nanoparticles"

_materials, 2023, doi:10.3390/ma16051973_

Round 1

Reviewer 1 Report

1.   There are some grammatical mistakes. Please carefully check the manuscript and the corrected language.

2.    Here author mention in introduction part “Upconversion devices (UCDs)” please explains in detail.

3.   In Figure 2, the FESEM image is blurry. To make them easier to read, I believe that they need to be made clearer.

4.   Different methods are applied for characterizing the as-prepared sample. However, the authors use these methods to prove some as-known results.

5.   The authors do not go into much detail about the results in the PL spectra and absorption spectra, so please explain them. 

6.   More information, such as XRD spectra for structural analysis, should be provided to describe the Ag Nanoparticles as-prepared.

Author Response

Thank you for your comments.

Please see the attachment about my response.

Reviewer 2 Report

In this work, Zhang et al. studied the upconversion properties of complex evices made of ITO/ Zn0.95Mg0.05O/ Ag NPs/ PbS QDs/ MoOx/ poly-TPD/ (CdSe/ZnS) QDs/ Zn0.95Mg0.05O/ Ag layers. The focus of the publication is to study the influence of the Ag surface plasmon on the photoelectric properties.

Overall, the novelty of the paper and the value of using Ag particles is not very clear.

The introduction is really too short and filled with generic sentences like "The Ag NPs have been found to have good optical and electrical properties in previous experiments.” Sometimes, the sentences are too vague: “However, due to the lattice mismatch, the inability to prepare in large area, and the type of substrate,[9]  other device structures have been considered”. The introduction needs to be reworked to bring more clarity, it needs to be more detailed and it needs to explain the state of the art concerning the integration of plasmonic materials in these devices.

The same problem is encountered later in the results section where, despite the fact that the publication focuses on the influence of Ag nanoparticles, the results are not compared with the same device but without Ag NPs. It is written “In the past research, the current of UCD without Ag NPs can only reach 10 A/m2, but in this study, it reaches to 2500 A/m2 “ but there is no reference to previous publication with the dame devices. Some detailed comparisons with and without Ag NPs need to be provided.

Finally, more structural data needs to be provided for both the Ag NPs and the whole device. In particular because the plasmonic response is strongly influenced by the shape and size of Ag NPs. Higher TEM resolution should be provided for the Ag NPs in order to determine the size of the NPs and their crystallinity. Finally, the whole device should be shown. In this regards, TEM cross-section images of the different layers as prepared by FIB would help to see the different layers and their respective thicknesses.  

Author Response

(The authors gave the same response as above.)

Reviewer 3 Report

In this work, quantum tunneling in the device was demonstrated and  the enhancement caused by localized surface plasmon was presented. While the authors have fabricated an interesting UCD, there are significant concerns that must be addressed before this manuscript is acceptable for publication.

·        Lines27-28: The author mentions “complicated steps” and “lattice mismatch” in the introduction paragraph, but without a little more information this is not much of an introduction. I recommend elaborating on this.

·        Lines 108-109: Five drops? All measurements had been so exact. Perhaps a volume is better here?

·        Figure 1:

o   Inset axis labels too small (not legible) for panels (a) and (c)

o   Perhaps I missed it - what is excitation for Fig. 1b & d PL?

o   Fig. 1 b & c not discussed in text at all

·        Figure 2: Did you use an element known to not be in your structure for a zero baseline? (in other words, to prove the EDX is showing what you want to show and not noise or different elemental compositions that are close in energy)

·        Line 148: Typo? “USD” à UCD (also in Fig. 3 caption)

·        Lines 150-151 à define acronyms at first instance of usage, even commonly used acronyms such as HOMO and LUMO

·        Minor comment: I recommend changing colors in Fig. 3 to make (a) and (b) self-consistent, i.e., some layers have colors in (a) and then a different color in (b), such as Ag NP

·        Fig. 4: What is the barrier with the Ag nanoparticles?

·        Fig. 5: Can the author comment on the asymmetry of the simulated electric field in the vicinity of the Ag islands? Why are the E-field magnitudes absent? Are the authors confident this is not due to meshing?

·        Can the authors comment on the differences they expect for silver square islands in Fig. 5 versus nanoparticles used in this work?

·        Line 221: Give citation (and more details) about the 10 A/m^2 measurement. An increase of 250x is large, and this reviewer would like some more details to generate some confidence in this.

·        Line 222: “Testing area of 4 mm” à presumably 4 mm^2?

Author Response

(The authors gave the same response as above.)

Round 2

Reviewer 2 Report

My comments were correctly adressed. The addition of the TEM micrograph and the redesign of the model is interesting. 

Reviewer 3 Report

My comments have been adequately addressed.